# Ginsenoside Rc Promotes Bone Formation in Ovariectomy-Induced Osteoporosis In Vivo and Osteogenic Differentiation In Vitro

**DOI:** 10.3390/ijms23116187

**Published:** 2022-05-31

**Authors:** Nan Yang, Xiao Zhang, Lingfeng Li, Tongtong Xu, Meihui Li, Qi Zhao, Jinling Yu, Jue Wang, Zhihui Liu

**Affiliations:** Department of Stomatology, Jilin University, Changchun 130000, China; yangnan19@mails.jlu.edu.cn (N.Y.); xiaozhang19@mails.jlu.edu.cn (X.Z.); lilf19@mails.jlu.edu.cn (L.L.); xtt19@mails.jlu.edu.cn (T.X.); limh19@mails.jlu.edu.cn (M.L.); zhaoqi20@mails.jlu.edu.cn (Q.Z.); yujl20@mails.jlu.edu.cn (J.Y.); wangjue19@mails.jlu.edu.cn (J.W.)

**Keywords:** bone formation, ginsenoside Rc, osteoporosis, Wnt signaling pathway

## Abstract

Ginsenoside Rc is one of the active components used in traditional Chinese medicine. We aim to explore how ginsenoside Rc can be used in the treatment of osteoporosis. Micro-CT demonstrated that the treatment of ovariectomized (OVX) mice with ginsenoside Rc significantly inhibited the decrease in bone mineral density, bone volumetric fraction, and trabecular number, and the increase in trabecular separation. Histological staining, qRT-PCR, and Western blot demonstrated that ginsenoside Rc enhances the microstructure of trabecular bone, and promotes the expression of bone formation-related genes. Alkaline phosphatase (ALP) staining, Alizarin Red staining, qRT-PCR, and Western blotting demonstrated that ginsenoside Rc dose-dependently promoted the osteogenic differentiation of MC3T3-E1 cells. A ginsenoside Rc-induced increase in the expression of β-catenin, p-GSK-3β, collagen-1, ALP, and RUNX-2 family transcription factor-2 was significantly attenuated upon 10 μM XAV-939 treatment, while the decrease in the expression of GSK-3β and p-β-catenin was significantly enhanced. Ginsenoside Rc promotes bone formation in ovariectomy-induced osteoporosis in vivo and promotes osteogenic differentiation in vitro via the Wnt/β-catenin signaling pathway.

## 1. Introduction

Osteoporosis is characterized by a significant reduction in bone mass, mineral density, and strength, leading to increased fragility and bone fractures with high morbidity and mortality [1]. Bone remodeling includes bone resorption and bone formation, and bone homeostasis is maintained through the dynamic balance of the two. In osteoporosis, bone formation is weakened and bone resorption is enhanced; therefore, bone resorption exceeds bone formation, resulting in changes in the quantity and structure of cortical and cancellous bone [2]. This further leads to a decrease in bone density, strength, and stiffness [3].

At present, the main clinical treatment for osteoporosis is drug-based interventions. Estrogen replacement treatment was once considered to be the first-line drug for the prevention and treatment of menopausal osteoporosis [4]. Estrogen can effectively inhibit bone resorption and regulate balance. However, in addition to acting on bone, estrogen can also strongly stimulate other tissues, causing breast cancer and uterine bleeding [5]. In a study [6] of breast cancer caused by long-term use of estrogen (average 17.2 years), 11.2% of estrogen users developed breast cancer, and the relative risk (RR) of estrogen use was 2.8 [95% confidence interval (95%CI) 1.3–5.9]. Bisphosphonates are currently commonly used drugs in the clinical treatment of osteoporosis, but the most serious side effects are atypical femoral fractures and jaw osteonecrosis. Therefore, finding a drug with fewer side effects and good effects has become an urgent problem to be solved in the treatment of osteoporosis. Therefore, more attention has been given to natural medicinal plants, especially some edible herbs and isolated active compounds, to treat this ailment.

Ginsenoside Rc, one of the major protopanaxadiol-type saponins, is one of the main active ingredients of traditional Chinese medicine. The chemical structure is shown in Figure 1.

Ginsenosides are the main active components of ginseng [7], which is a rare and precious Chinese medicine produced in Northeast China. Traditional Chinese medicine is famous for its multiple pharmacological effects and mild side effects [8,9].

Ginsenoside Rc has analgesic, anti-allergy, anti-tumor, and sedative effects [10]. Ginsenoside Rc may be a potent antidiabetic drug that significantly enhances glucose uptake [11]. It exhibits potent anti-inflammatory activities both in vitro and in vivo [12,13]. In addition, ginsenoside Rc is involved in scavenging reactive oxygen species (ROS) [14].

In recent years, a large number of studies have revealed that ginsenosides can promote cell proliferation and osteogenesis, as well as inhibit osteoclast activity. The protopanaxadiol-type ginsenosides include Rb1, Rb2, Rb3, Rc, Rd, and Rh2. Among them, ginsenosides Rb1 [15], Rb2 [16], Rd [17], and Rh2 [18,19] promote cell proliferation and osteogenic differentiation. The chemical structure of Rc is similar to that of these. Compound K (CK), the metabolite of Rc, can promote osteogenesis through the Wnt signaling pathway [20].

Genetic studies in both humans and mice have established that Wnt signaling is crucial in stimulating osteoblastogenesis and the bone formation process [21,22,23]. Accordingly, this signaling is currently an enticing target for drug development to battle skeletal diseases [24].

Therefore, we wondered whether ginsenoside Rc could be used to treat osteoporosis via activating the Wnt signaling pathway. In the present study, we aim to investigate the effects of ginsenoside Rc on osteoporosis and the underlying mechanisms in vitro and in vivo.

The most commonly used model for the establishment of osteoporosis is the ovariectomized model (OVX) obtained by ovariectomy. Following ovariectomy, animals develop symptoms similar to the state of postmenopausal bone loss in humans [25]. In the in vivo study, we chose C57Bl/6 female SPF mice to establish the OVX model, which has the advantages of being inexpensive and less demanding on feeding, while C57Bl/6 is easy to reproduce and robust, and is widely used in physiological research. MC3T3-E1 cells were selected for in vitro experiments in this study. The MC3T3-E1 mouse calvaria-derived preosteoblast cell line established in Japan [26] has contributed remarkably to the investigation of the role of osteoblasts in bone formation [27,28].

## 2. Results

### 2.1. Ginsenoside Rc Promotes Bone Formation in Ovariectomy-Induced Osteoporosis In Vivo

#### 2.1.1. Ovariectomized Mice Have Increased Weight

An animal model of osteoporosis was successfully established. To explore the effect of ginsenoside Rc, OVX or sham-operated mice were administered 25 mg/kg ginsenoside Rc, 50 mg/kg ginsenoside Rc, 1 mg/kg 17 β-estradiol, or NaCl via gavage every day for 3 months. There were no animal deaths or other adverse events prior to the sacrifice. Weekly body weight measurements demonstrated that mice in the OVX group were significantly heavier than the sham group from week 4. The body weight of mice in the OVX + 25 mg/kg Rc, OVX + 50 mg/kg Rc, and OVX + E2 groups was significantly lower than that in the OVX group (Figure 2A). No obvious pathological changes in the heart, liver, spleen, lungs, and kidneys were observed (Figure 2B).

#### 2.1.2. Ginsenoside Rc Can Increase BMD in OVX Mice

Micro-computed tomography (micro-CT) exhibited a 14.2% reduction in bone mineral density (BMD), a 68.4% reduction in bone volume fraction (BV/TV), a 57.4% reduction in the number of trabeculae (Tb.N), and a 153% increase in trabecular separation (Tb.Sp) in the OVX group compared with the sham group (Figure 3A,B). Only BMD was elevated in the OVX + 25 mg/kg Rc group compared with the OVX group. Compared with the OVX group, the BMD, BV/TV, and Tb.N of the OVX + 50 mg/kg Rc group and the OVX + E2 group were all increased, and the Tb.Sp were all decreased. Although ginsenoside Rc treatment did not completely protect mice from OVX-induced bone loss, bone mass in OVX mice treated with 50 mg/kg ginsenoside Rc almost recovered the parameters to pre-surgery values.

#### 2.1.3. Effects of Ginsenoside Rc on Trabecular Bone Tissue of Distal Femur in Mice

Histological analysis of the distal femur was performed using hematoxylin and eosin (HE) staining, Masson staining, and immunohistochemical staining (Figure 4A,B). HE staining showed that compared with the sham group, the distal femur volume of the OVX group was larger, and the trabecular arrangement of the metaphysis was relatively disordered. The number of trabecular bones was increased in the OVX + 25 mg/kg Rc group compared to the OVX group. The volume of the distal femur in the OVX + 50 mg/kg Rc group and the OVX + E2 group was decreased, and the metaphysis trabecular arrangement was relatively orderly. Masson staining demonstrated that the blue collagen fibers were greatly reduced in the OVX group compared with the sham group. Compared with the OVX group, OVX + 25 mg/kg Rc, OVX + 50 mg/kg Rc, and OVX + E2 groups exhibited a large number of blue collagen fibers in the growth plate and trabecular bone. Immunohistochemistry demonstrated very weak staining of alkaline phosphatase (ALP), bone morphogenetic protein (BMP)-2, RUNX family transcription factor 2 (RUNX-2), collagen-1 (COL-1), and osteocalcin (OCN) in the OVX group compared to the sham group. Compared with the OVX group, the staining of OVX + 25 mg/kg Rc, OVX + 50 mg/kg Rc, and OVX + E2 groups was significantly enhanced.

#### 2.1.4. Ginsenoside Rc Prevents OVX-Induced Bone Loss by Promoting the Expression of Bone Formation-Related Genes

The *Col1, Alp, Runx2, Bmp2,* and *Ocn* mRNA expression in the OVX group was significantly lower than those in the sham group (Figure 5A), and the mRNA expression of *Alp, Runx2, Bmp2* in the OVX + 25 mg/kg Rc group was significantly increased. Furthermore, *Col1, Alp, Runx2, Bmp2,* and *Ocn* mRNA expression was significantly increased in the OVX + 50 mg/kg Rc group. In terms of mRNA expression, the OVX + 50 mg/kg Rc group was second only to the OVX + E2 group. The expression of *Col1, Alp, Runx2, Bmp2,* and *Ocn* in the OVX group at the protein level was consistent with mRNA expression results and significantly lower than that in the sham group (Figure 5B). There was no significant difference between the OVX + 25 mg/kg Rc group and the OVX group in terms of protein expression. However, the expression of COL-1, ALP, RUNX-2, BMP-2, and OCN was significantly increased in the OVX + 50 mg/kg Rc group, and the therapeutic effect was comparable to that in the OVX + E2 group. Therefore, ginsenoside Rc prevented OVX-induced bone loss by promoting the expression of bone formation-related genes.

### 2.2. Ginsenoside Rc Promotes Osteogenic Differentiation In Vitro

#### 2.2.1. Low Concentrations of Ginsenoside Rc Promote the Viability of MC3T3-E1 Cells

We used cell counting kit-8 (CCK-8) to evaluate the effect of ginsenoside Rc on MC3T3-E1 cell viability. MC3T3-E1 cells grew well (Figure 6A). MC3T3-E1 cells were cultured with different concentrations (0, 25, 50, 100, 200, 400, and 800 μM) of ginsenoside Rc for 1, 3, and 5 days (Figure 6B). The results demonstrated that 25–200 μM ginsenoside Rc exhibited the effect of promoting cell viability on the first day. On the third day, 50–100 μM ginsenoside Rc exhibited the effect of promoting cell viability, while 800 μM ginsenoside Rc exhibited significant inhibition of cell viability. On the fifth day, only 25 μM ginsenoside Rc promoted cell viability, while 100–800 μM ginsenoside Rc exhibited significant inhibition of cell viability. To determine whether high concentrations of ginsenoside Rc were toxic to cells after day 5 and whether high concentrations of ginsenoside Rc promoted cell differentiation and thus slowed cell viability, we performed live-dead cell staining. Live-dead cell staining (Figure 6C) and cell count analysis (Figure 6D) demonstrated that the proportion of dead cells upon treatment with 100–200 μM ginsenoside Rc was significantly increased. Therefore, 100–200 μM ginsenoside Rc was not conducive to cell survival after day 5.

#### 2.2.2. Ginsenoside Rc Dose-Dependently Promotes the Osteogenic Differentiation of MC3T3-E1 Cells

ALP activity was enhanced in MC3T3-E1 cells cultured in an osteogenic induction medium for 7 and 14 days. ALP activity was increased in a dose-dependent manner upon treatment with 0–200 μM of ginsenoside Rc (Figure 7A,B).

Alizarin Red S staining exhibited an intense red color in MC3T3-E1 cells after 21 days of differentiation (Figure 8A). Consistent with the staining results, the Alizarin Red signal quantification demonstrated that 0–200 μM ginsenoside Rc dose-dependently promoted calcium deposition (Figure 8B).

#### 2.2.3. Ginsenoside Rc Promotes Expression of Osteogenesis-Related Genes

Based on the results of CCK-8, live-dead cell staining, ALP staining, and Alizarin Red S staining, we believe that 50 μM ginsenoside Rc was appropriate to promote osteogenic differentiation of MC3T3-E1 cells. After cells were treated with 50 μM ginsenoside Rc, the expression of *Col1, Alp, Runx2, Bmp2,* and *Ocn* mRNA was 11–26 times higher than that in the control group after 14 days (Figure 9A). The expression of osteogenesis-related proteins in the treatment group was about 1.2 times that of the control group (Figure 9B,C).

#### 2.2.4. Ginsenoside Rc Promotes Osteogenic Differentiation of MC3T3-E1 Cells via the Wnt/β-Catenin Signaling Pathway

To explore the pathway by which ginsenoside Rc regulated the osteogenic differentiation of MC3T3-E1 cells, we examined the Wnt/β-catenin signaling pathway, which is crucial in osteogenesis. The expression of relevant proteins was investigated via Western blotting on day 14 of osteogenesis. XAV-939 was used to further verify whether the Wnt/β-catenin pathway is involved in ginsenoside Rc-induced osteogenic differentiation. According to different groups, 50 μM ginsenoside Rc and/or 10 μM XAV-939 were added to the culture medium. Increased expression of β-catenin, p-GSK-3β, COL-1, ALP, and RUNX-2, and decreased expression of p-β-catenin and GSK-3β were observed in the 50 μM ginsenoside Rc-treated group (Figure 10A,B). However, the ginsenoside Rc-induced increase in the expression of β-catenin, p-GSK-3β, COL-1, ALP, and RUNX-2 was significantly attenuated upon treatment with 10 μM XAV-939, while the reduction of p-β-catenin and GSK-3β expression were significantly enhanced. The expression levels were normalized relative to GAPDH. The results indicated that the Wnt/β-catenin signaling pathway may be involved in the osteogenic differentiation of MC3T3-E1 cells promoted by ginsenoside Rc.

## 3. Discussion

Recently, more attention has been given to natural medicinal plants, especially some edible herbs and isolated active compounds, to treat osteoporosis, due to the nature of the disease and side effects of current treatments [29,30,31].

Ginsenoside Rc is one of the active ingredients of many traditional Chinese medicines and has a variety of pharmacological effects. In terms of anti-arthritis, ginsenoside Rc (40 μg/mL and 60 μg/mL) exhibits significant activity, which can alleviate inflammation by inhibiting the targeted binding kinase 1/interfering regulatory factor-3 and p38/activating transcription factor-2 pathways [13]. Lee et al. [32] found that ginsenoside Rc had the potential to prevent the breakdown of cartilage collagen matrix and inhibit MMP-13 expression in human chondrocytes treated with IL-1β at non-cytotoxic concentrations (1–50 μmol/L). Ginsenoside Rc has a significant anti-oxidative effect. Sirtuin 1 (SIRT1) belongs to the family of NAD+-dependent histone deacetylases, and SIRT1 plays a key role in the oxidative stress response. Wang et al. [33] found that ginsenoside Rc could enhance the acetylation activity of SIRT1, thereby inhibiting the formation of intracellular ROS. ROS can increase bone resorption by enhancing the development and activity of osteoclasts [34].

Lee et al. [35] used chromatography to verify that the main part of the ginseng water extract included 1.19% Rb1, 0.12% Rb2, 0.57% Rg1, 0.07% Rc, 0.64% Re, and 0.04% Rf. It was found that in OVX rats treated with ginseng water extract (100, 200, 300, or 500 mg/kg), the sharp decrease in BMD and the deterioration of trabecular bone structures can be significantly reduced. In the presence of ginseng water extract, other bone remodeling markers such as Tb.N, Tb.T, and Tb.Sp were also significantly restored. They also exposed RAW 264.7 cells to RANKL and M-CSF receptor activator for 5 days, and each group was added with ginseng water extract (0.05, 0.1, 0.25, 0.5, or 1 µg/mL). It was found that Rb1, Rg1, Rc, Re, and Rf can inhibit osteoclast differentiation, TRAP activity, and staining.

Given that the detailed anti-osteoporosis activity of ginsenoside Rc is still unclear, the present study was designed to systematically evaluate the therapeutic effect of ginsenoside Rc as well as underlying molecular mechanisms.

In the in vivo experiment, the dose of Rc is 25 mg/kg or 50 mg/kg, the injection volume is 0.4 mL, each mouse weighs about 20 g, and the concentrations are about 1.25 mg/mL and 2.5 mg/mL, respectively. In the in vitro experiment part, the converted concentration is about 0.054 mg/mL, calculated according to the Rc concentration of 50 μM. Although the Rc concentrations in vivo are about 25–50 times higher than that in vitro, it needs to be taken into account that these Rc concentrations might still be higher than those concentrations detected in human and animal plasma. According to a former pharmacokinetic study [10], the bioavailability of Rc is relatively low, which is only about 0.17%. Considering that the concentration in plasma will decrease significantly after oral administration, the above-mentioned concentrations were selected.

As expected, our research demonstrated that ginsenoside Rc could enhance bone mineral density as well as improve the microarchitecture of bone trabecular, and prevent bone loss in osteoporotic mice induced by ovariectomy. There is no doubt that estrogen plays an important role in stimulating bone metabolism, and postmenopausal osteoporosis, a sex-steroid deficiency state characterized by decreased bone mineral density and increased risk of fracture, is believed to be mainly caused by estrogen deficiency [36]. Ovariectomy could lead to the imbalance of bone turnover, causing higher bone resorption and lower bone formation, and eventually inducing osteoporosis [37].

According to previous data from the literature, after OVX surgery and during postmenopausal osteoporosis, the body weight is increased [38]. In the present study, treatment with estradiol and ginsenoside Rc significantly reversed the body weight gain (Figure 2A). This suggests that ginsenoside Rc may reverse weight gain due to estrogen deficiency through phytoestrogens.

The micro-CT technology can give us both intuitive and quantitative data on the microarchitecture of trabecular bone. The data obtained from the micro-CT is more meaningful and comprehensive for the diagnosis of osteoporosis. In the present study, BMD, TV/BV, and Tb.N were significantly ameliorated by ginsenoside Rc, whereas the increase in Tb.Sp was reversed (Figure 3A,B). Therefore, our data implied a promoting effect of ginsenoside Rc on bone formation in OVX mice.

Histological staining demonstrated that the distal femur volume in the OVX group was larger, and the metaphyseal trabecular arrangement was relatively disordered, as compared with the sham group. The number of trabeculae in the ginsenoside Rc treatment group was higher and the arrangement was more orderly, compared with the OVX group (Figure 4A,B). This indicated that ginsenoside Rc could prevent bone loss and improve bone structure in OVX mice. Immunohistochemical staining demonstrated that ALP, BMP-2, RUNX-2, COL-1, and OCN in the OVX group were significantly attenuated compared with the sham group. In addition, ALP, BMP-2, RUNX-2, COL-1, and OCN in the OVX+50 mg/kg Rc group and the OVX+25 mg/kg Rc group were significantly higher than those in the OVX group. QRT-PCR and Western blotting further verified the results of immunohistochemistry. The expression levels of *Col1*, *Alp*, *Runx2*, *Bmp2*, and *O*cn at both mRNA and protein levels in the OVX+50 mg/kg Rc group were significantly higher than those in the OVX group, and the efficacy was comparable to the OVX+E2 group (Figure 5A,B).

The expression of RUNX-2 indicates the onset of osteogenic differentiation, which occurs at the earliest stage of bone formation [39]. ALP, as an essential enzyme, also appears in the early stage of osteogenic differentiation [40]. OCN is a non-collagenous protein in the bone matrix that is secreted by osteoblasts during mineral formation [41]. In addition, OCN is closely related to hydroxyapatite in bone tissue and may be involved in regulating calcium deposition. COL-1 is one of the extracellular matrix proteins secreted by osteoblasts, which can serve as a structural framework for the maturation of the extracellular matrix and the formation of calcified nodules [42,43,44]. BMP-2 belongs to the BMP family; in addition to directly promoting bone formation, it can also induce the generation of ALP through various pathways [45]. Genes encoding all five proteins were upregulated, indicating that ginsenoside Rc played an important role in the early and late stages of osteoblast differentiation. From the perspective of histology and molecular biology, we demonstrated that ginsenoside Rc could promote the bone formation process in OVX mice by promoting the expression of bone formation-related genes.

To our knowledge, this is the first study demonstrating that ginsenoside Rc promotes the osteogenic differentiation of MC3T3-E1 cells by regulating the Wnt/β-catenin signaling pathway. Based on the results of CCK-8, live-dead cell staining, ALP staining, and Alizarin Red S staining, we believe that 50 μM ginsenoside Rc is an appropriate concentration to promote osteogenic differentiation of MC3T3-E1 cells. Ginsenoside Rc at 50 μM promoted the expression of osteogenesis-related genes, including *Col1, Alp, Runx2, Bmp2,* and *Ocn* (Figure 9A–C).

Wnt proteins are members of a family of secreted molecules involved in promoting osteoblast function [46,47]. As a basic signaling pathway in bone formation, the Wnt signaling pathway includes two main pathways: The canonical Wnt pathway and the non-canonical Wnt pathway. In the canonical Wnt pathway, when the Wnt signaling pathway is inactive, β-catenin is phosphorylated by Glycogen synthase kinase 3β (GSK-3β), which in turn activates the ubiquitin system, leading to its degradation through the proteasomal pathway [48,49,50,51,52]. As shown in Figure 11, when proteins, such as Wnt3a, Wnt8, and Wnt10b, bind to the Frizzled (Fz) receptor, Dishevelled (DSH) will be generated, and then the signal will be transmitted to GSK-3, Adenomatosis Polposis Colis (APC), and Axin, so that their inhibitory effect on β-catenin is weakened [53]. In the nucleus, β-catenin binds to the intranuclear transcription factor, T-Cell factor/Lymphoid Enhance Factor (TCF/LEF), and promotes the transcription of its target genes [54]. XAV-939 has been widely used and shown to be an inhibitor of the Wnt signaling pathway [55]. Recent studies have demonstrated that XAV-939 can inhibit osteogenesis by inhibiting the canonical Wnt signaling pathway [56]. We explored the mechanism of how ginsenoside Rc promotes the osteogenic differentiation of MC3T3-E1 cells. Compared with the ginsenoside Rc(−) XAV-939(−) group, the expression of β-catenin, p-GSK-3β, COL-1, ALP, and RUNX-2 was increased in the ginsenoside Rc (+) XAV-939(−) group; while the expression of p-β-catenin and GSK-3β was decreased. As described above, Wnt signaling is activated by osteoblasts, regulates the expression of bone marker-related genes, such as *RUNX2* and *ALP*, and promotes osteoblast viability and osteogenic activity. GSK-3β can degrade β-catenin, so the expression of GSK-3β shows the opposite trend to that of β-catenin. Therefore, ginsenoside Rc can effectively activate the Wnt signaling pathway at a concentration of 50 μM. In the ginsenoside Rc (+) XAV-939 (+) group, the upregulation of β-catenin, p-GSK-3β, COL-1, ALP, and RUNX-2 induced by ginsenoside Rc was significantly attenuated after treatment with 10 μM XAV-939; while the downregulation of p-β-catenin and GSK-3β expression was significantly enhanced (Figure 10A,B). This indicates that XAV-939 can inhibit the Wnt signaling pathway activated by ginsenoside Rc, as shown in Figure 11.

## 4. Materials and Methods

### 4.1. Ovariectomy-Induced Osteoporosis Mice Model

All mice-related experiments were carried out in the Laboratory Animal Center of Jilin University. All experiments were reviewed and approved by the Institutional Animal Care and Use Committee and complied with the requirements of Jilin University and the state for the ethical welfare of experimental animals. The breeding conditions were controlled in strict accordance with GB14925. The light time is 8:00–20:00, the temperature is maintained at 23 ± 1 °C, and the humidity is controlled at 50–60%. The animals are guaranteed to have sufficient food and drinking water and can eat freely. The feed is a maintenance feed for rats and mice. The main ingredients include corn, soybean meal, fish meal, flour, bran, salt, calcium chloride phosphate, stone powder, multivitamins, various trace elements, amino acids, etc.

Sixty 6–8 weeks old C57Bl/6 female SPF mice were purchased from Liaoning Changsheng Experimental Animal Co., Ltd., and each mouse weighed about 20 g. The conditions were as follows: Light between 8:00–20:00, the temperature was maintained at 23 ± 1 °C, and humidity was regulated at 50–60%. The animals had sufficient food and drinking water.

The mice were randomly divided into five groups as follows: Sham control group (sham); OVX + NaCl (OVX); OVX + 25 mg/kg ginsenoside Rc (OVX + 25 mg/kg Rc); OVX + 50 mg/kg ginsenoside Rc (OVX + 50 mg/kg Rc); OVX + 1 mg/kg 17 β-estradiol (OVX + E2).

Mice in the OVX and OVX drug treatment groups were anesthetized with 1% sodium pentobarbital at 50 mg/kg and then underwent ovariectomy. The surgical procedure is shown in Figure 12. After incising the skin and muscle through back costal ridge incision and separating the cellulite, the pink ovaries were visible. The ovaries were clamped with hemostatic forceps, then the fallopian tubes (including fat) were ligated under the ovaries, the ovaries were cut, and the skin and muscles were sutured in layers. In the sham operation group, adipose tissue about the same mass as the ovary was removed near the ovary.

The animals in each group were given 25 mg/kg ginsenoside Rc, 50 mg/kg ginsenoside Rc, 1 mg/kg 17 β-estradiol or NaCl via gavage 7 days after ovaries removal. The drug was administered every day for 3 months. Bodyweight was measured every week. After the last gavage, the animals were sacrificed by cervical dislocation. The femur, heart, liver, spleen, lung, and kidney of each mouse were collected. After the experiment, the animal carcasses were disposed of safely.

### 4.2. Cell Culture and Differentiation

The mouse osteoblastic cell line MC3T3-E1 was obtained from Zhongqiaoxinzhou (Shanghai, China). Cells were cultured in α-MEM (Gibco, Grand Island, NY, USA) with 10% fetal bovine serum (BI, Kibbutz Beit Haemek, Israel) and 1% penicillin-streptomycin at 37 °C.

To induce differentiation, the cells were seeded in a six-well plate at 40,000 cells/well. The next day, the original medium was discarded, and the culture medium was replaced with an osteogenic induction medium which included 50 mg/mL L-Ascorbic acid (Sigma-Aldrich, St.Louis, MO, USA) and 1 M β-glycerophosphate sodium (Solarbio, Beijing, China), and the fetal bovine serum concentration was 5%. Ginsenoside Rc (Yuanye, Shanghai, China) (0, 25, 50, 100, 200 μM) also needs to be added to the osteogenic induction medium. The osteogenic induction medium was changed every 2–3 days. Ginsenoside Rc was incubated with cells continuously for 14–21 days.

### 4.3. Micro-CT

The microarchitecture of the 15 femoral trabecular bones was investigated using micro-CT (SCANCOµCT; Skyscan1172), performed at a resolution of 20 µm. Image reconstruction was performed using the relevant measurement software with a protocol library. Reconstructed cross-sections were reorientated, and the region of interest was further selected. The volume of interest was 1–1.5 mm below the growth plate. The trabecular metric parameters measured included trabecular BMD, BV/TV, Tb.Sp, and Tb.N.

### 4.4. Histological Analysis of OVX Mice

After micro-CT, the femur was decalcified with 15% EDTA for 3 months. After gradient dehydration and paraffin embedding, the femur was cut into 4 mm sections for further analysis, including HE staining, Masson staining, and immunohistochemistry.

For immunohistochemical staining, the paraffin sections were baked at 60 °C, then dewaxed in xylene, then dehydrated with gradient ethanol, and retrieved at 37 °C for 30 min. Endogenous peroxidase activity blocking and serum blocking was performed using an immunohistochemical staining kit (MXB, Fuzhou, China). Primary antibodies against ALP (Proteintech 13365-1-AP, 1:400, Wuhan, China), BMP-2(Affinity, AF5163, 1:300, Jiangsu, China), RUNX-2 (Abways, CY5395, 1:800, Shanghai, China), OCN (Biorbyt, orb348959, 1:300, Cambs, UK), and COL-1(Proteintech, 66761-1-Ig, 1:4800, Wuhan, China) were diluted in proportion and added dropwise to the tissue, and incubated at 4 °C overnight. The next day, after incubation with biotin-labeled secondary antibody and streptomyces antibiotic-peroxidase solution, DAB (MXB, China) was used for color development, and the positive expression was specific brownish-yellow. The sections were imaged under a microscope (OLYMPUS, BX53F, Tokyo, Japan) and analyzed using OLYMPUS cellSens Entry 2.2 software. In the expression analysis of immunohistochemically stained sections of different groups, the same area and the same conditions were selected for gray density analysis using Image J software ( Image J 1.53c, Wayne Rasband, National Institutes of Health, Bethesda, ML, USA).

### 4.5. qRT-PCR

TRIeasy Total RNA Extraction reagent (Yeasen Biotech, Shanghai, China) was added to the metaphysis of mouse femur and homogenized with a homogenizer (JXFSTPRP-24, Shanghai Jingxin Industrial Development Co., Ltd., Shanghai, China) using the following parameters: three grinding times, a frequency of 73 Hz, an interruption time of 3 s, and a running time of 60 s. The cDNA was synthesized using the Hifair III 1st strand cDNA Synthesis SuperMix for qPCR (Yeasen Biotech, Shanghai, China). Target genes were amplified using CFX Connect Real-Time PCR Detection System (Bio-Rad, Singapore) with Hifair qPCR SYBR Green Master Mix (Yeasen Biotech, Shanghai, China). The primer sequences used are shown in Table 1.

qRT-PCR of the mouse osteoblastic cell line MC3T3-E1 was also performed as described above using the same reagents.

### 4.6. Western Blotting

RIPA buffer (Beyotime, Shanghai, China) containing 1% PMSF (Beyotime, Shanghai, China) was added to the metaphysis of mouse femur and homogenized with a homogenizer. After centrifugation at 12,000 rpm, the supernatant was collected and the protein concentration was measured using the BCA assay kit (Beyotime, Shanghai, China). The protein sample was boiled at 100 °C for 5 min. Exactly 10 μg of protein was separated via 10% SDS-polyacrylamide gel electrophoresis and transferred onto polyvinylidene difluoride (PVDF) membranes. Then, the membranes were blocked with BSA (Solarbio, Beijing, China) for 1 h and incubated with the primary antibodies against ALP (Proteintech 13365-1-AP, 1:1000, Wuhan, China), RUNX2 (Abways, CY5395, 1:500, Shanghai, China), COLI (Proteintech, 66761-1-Ig, 1:2000, Wuhan, China), Beta-Catenin (Proteintech, 51067-2-AP, 1:5000, Wuhan, China), Phospho-Beta-Catenin (Proteintech, 80084-1-RR, 1:5000, Wuhan, China), GSK-3α/β (CST, 5676, 1:1000, Danvers, MA, USA), Phospho-GSK-3α/β (CST, 8566, 1:1000, Danvers, MA, USA), BMP-2 (Affinity, AF5163, 1:500, Jiangsu, China), OCN (Biorbyt, orb348959, 1:500, Cambs, UK) and GAPDH (Abways, ab0037, 1:4000, Shanghai, China) overnight. The membranes were washed the membrane with TBST and then incubated with the secondary antibody (Proteintech SA00001-1, SA00001-2, 1:2000, Wuhan, China). Finally, the protein bands were visualized via enhanced chemiluminescence under a fully automatic chemiluminescence imager (Jun Yi Dong Fang, JY-MINI610, Beijing, China) and the gray values were analyzed using Image J software.

MC3T3-E1 cells were seeded onto a six-well plate at 40,000 cells/well. The next day, the culture medium was replaced with an osteogenic induction medium. When exploring whether ginsenoside Rc regulates the osteogenic differentiation of MC3T3-E1 cells through the Wnt/β-catenin signaling pathway, we used the following groups: An ordinary osteogenic induction medium [ginsenoside Rc(−) XAV-939(−)]; 50 μM ginsenoside Rc added to the osteogenic induction medium [ginsenoside Rc(+) XAV-939(−)]; 50 μM ginsenoside Rc and 10 μM XAV-939 added to the osteogenic induction medium [ginsenoside Rc (+) XAV-939 (+)]; 10 μM XAV-939 added to the osteogenic induction medium [ginsenoside Rc (−) XAV-939 (+)]. Proteins were extracted from cells after 14 days of incubation with RIPA buffer containing 1% PMSF and 1% phosphatase inhibitor (MCE, Shanghai, China) using a cell scraper. Western blotting was performed as described above.

### 4.7. Cell Viability Assay

To examine the effect of ginsenoside Rc on cell viability, MC3T3-E1 cells (1000 cells/well) were seeded onto a 96-well plate. On the second day, the medium was replaced with a medium containing different concentrations of ginsenoside Rc (0, 25, 50, 100, 200, 400, and 800 µM). After adding CCK-8 solution to each well on days 1, 3, and 5, the cells were incubated for 1 h at 37 °C, and then, the absorbance was measured at 450 nm using a microplate reader (SynergyHT, Bio Tek, Vermont, USA).

### 4.8. Live-Dead Cell Staining

MC3T3-E1 cells were cultured in six-well plates at a density of 40,000 cells/well. The next day, the original medium was discarded, and the medium containing different concentrations of ginsenoside Rc (0, 25, 50, 100, 200μM) was added for cell culture. Five days later, a Live-Dead Cell Staining Kit (calcein AM/PI double staining kit, Beyotime, Shanghai, China) was used for detecting the dead cells. Cells were washed with PBS and stained with calcein AM and PI. After incubation for 15 min at 37 °C, live and dead cells were visualized via fluorescence microscopy. Image J software was used to count live and dead cells separately, and then calculate the percentage of dead cells to total cells.

### 4.9. ALP Staining and Activity

The cells were seeded onto a six-well plate at 40,000/well. The next day, the culture medium was replaced with an osteogenic induction medium containing ginsenoside Rc (0, 25, 50, 100, 200 μM). The osteogenic induction medium also included 50 mg/mL L-Ascorbic acid (Sigma-Aldrich, St. Louis, MO, USA) and 1M β-glycerophosphate sodium (Solarbio, Beijing, China), and the fetal bovine serum (BI, Kibbutz Beit Haemek, Israel) concentration was 5%. After 14 days, the cells were fixed in 4% paraformaldehyde for 30 min and then washed thrice with ddH_2_O. Next, the cells were stained with the BCIP/NBT alkaline phosphatase color development kit (Beyotime, Shanghai, China). We prepared the lysate of RIPA and PMSF according to 100:1, collected the samples with a cell scraper, centrifuged them at 12,000 rpm, and collected the supernatant. The ALP assay kit (Nanjing Jiancheng Bioengineering Institute, Nanjing, China) combined with the Enhanced BCA Protein Assay Kit (Beyotime, Shanghai, China) was used to determine the ALP activity of each group of cells.

### 4.10. Alizarin Red S Staining

The cell seeding and osteoinduction were performed as described above. After 21 days, the cells were fixed, and Alizarin Red S solution (1%, pH 4.2, Solarbio, Beijing, China) was added. The staining was terminated after 30 min of incubation at 37 °C, and observed under the light microscope. Exactly 1 mL of 10% hexadecylpyridinium chloride monohydrate was added to each well. After incubating for 1 h at room temperature, the absorbance was measured at 562 nm using a microplate reader.

### 4.11. Statistical Analysis

Data were expressed as the mean ± standard deviation (SD) using GraphPad Prism 8.0.1. All experiments were performed independently at least three times. The normal distribution of the data was assessed using the Shapiro–Wilk test. For data normally distributed, a t-test was used for comparison between two groups, and one-way ANOVA was used to analyze significant differences among groups via SPSS software (SPSS 22.0.0.0, IBM, Armonk, NY, USA). The LSD post hoc test was then used for data with homogeneity of variance and Dunnett’s T3 test for the data with unequal variance. Nonparametric tests were used for not normally distributed data. When *p* < 0.05, the results were significantly different.

## 5. Conclusions

In conclusion, this study sought to identify the effects of ginsenoside Rc in the treatment of osteoporosis in cell and mouse models of osteoporosis. In vivo and in vitro data confirmed the beneficial roles of ginsenoside Rc against osteoporosis. In this study, an animal model of osteoporosis was established. From the perspectives of imaging, histology, and molecular biology, we confirmed that ginsenoside Rc can enhance the microstructure of trabecular bone, and promote the expression of bone formation-related genes. Ginsenoside Rc promotes osteoblast differentiation and matrix mineralization by activating the canonical Wnt pathway involving β-catenin and Runx2. Ginsenoside Rc increases bone formation by upregulating the expression of bone markers. Ginsenoside Rc is expected to be a natural substitute for the prevention and treatment of postmenopausal osteoporosis.

There is still room for further research in the future. This paper only reports the osteogenic effect of ginsenoside Rc, but osteoporosis is a joint effect of bone formation and bone resorption. The effect of ginsenoside Rc on bone resorption is worthy of further study. According to the role of ginsenoside Rc in promoting osteogenesis in this study, ginsenoside Rc can be loaded into bone scaffold materials in the future to provide potential cytokines for tissue-engineered bone and provide new ideas for the reconstruction of bone defects.

## Figures and Tables

**Figure 1 ijms-23-06187-f001:**
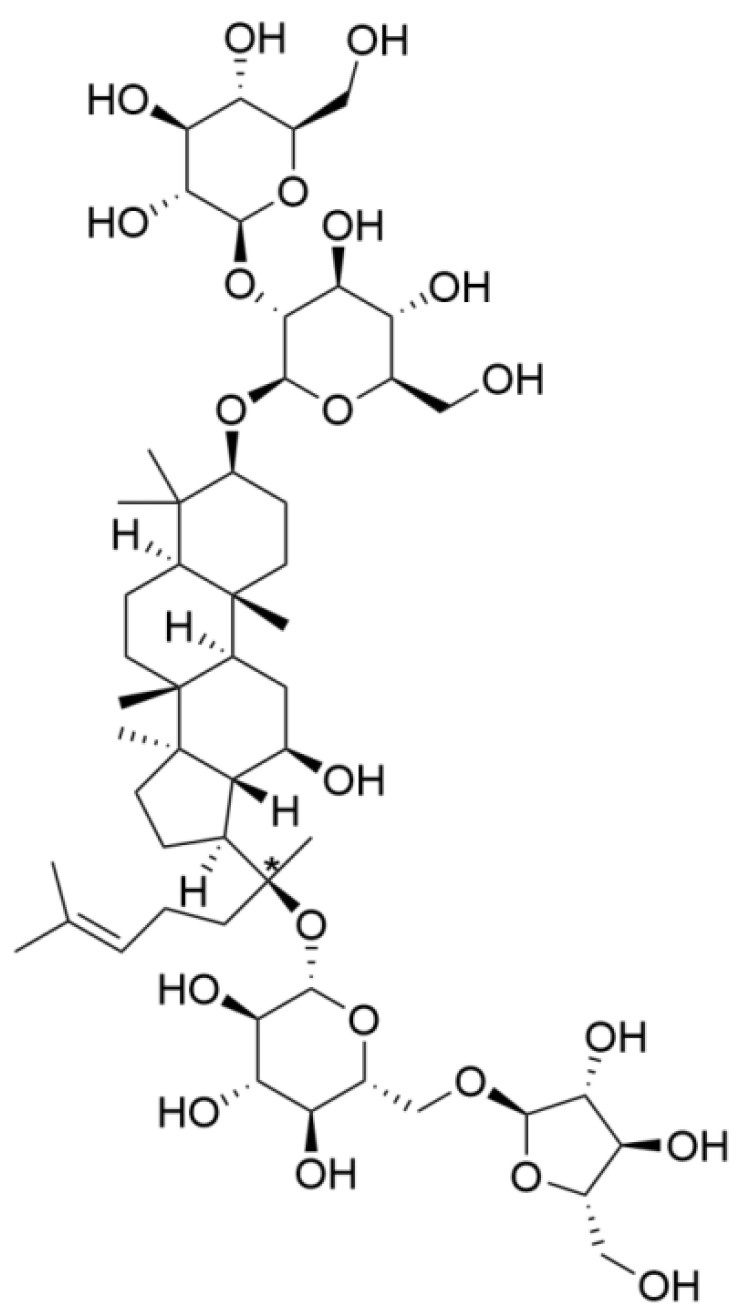
The chemical structure of ginsenoside Rc.

**Figure 2 ijms-23-06187-f002:**
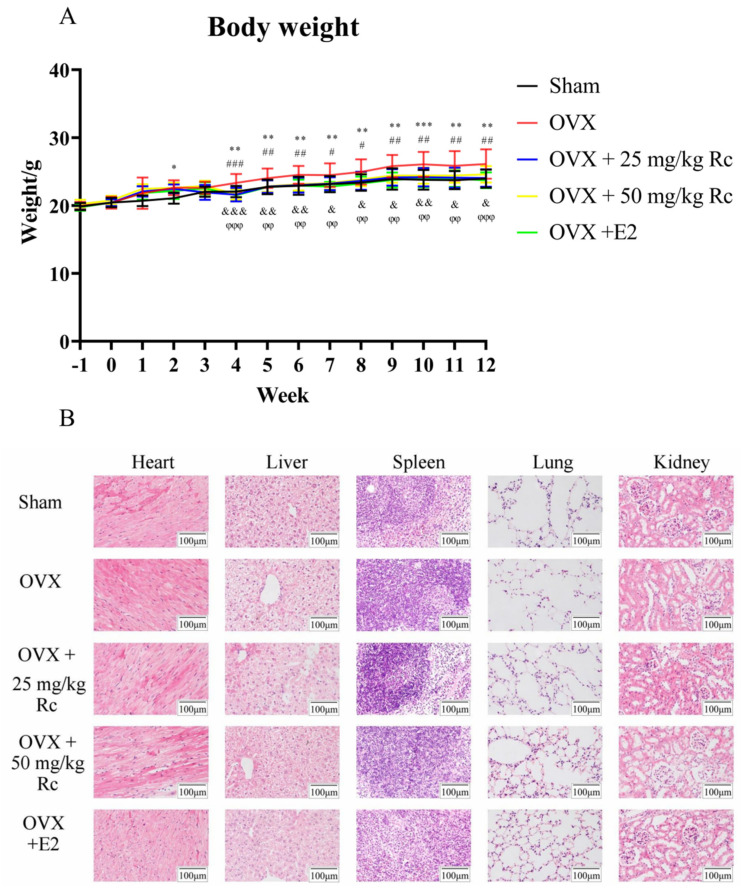
(**A**) Weight of mice in five groups. Week −1 represents the day when mice underwent ovariectomy. Week 0 represents the day that the administration of the therapeutic drug via gavage began. * *p* < 0.05, ** *p* < 0.01, *** *p* < 0.001, for the Sham group vs. OVX group. # *p* < 0.05, ## *p* < 0.01, ### *p* < 0.001, for the OVX + 25 mg/kg Rc group vs. OVX group. & *p* < 0.05, && *p* < 0.01, &&& *p* < 0.001, for the OVX + 50 mg/kg Rc group vs. OVX group. φφ *p* < 0.01, φφφ *p* < 0.001, for the OVX + E2 group vs. OVX group. (**B**) HE staining of heart, liver, spleen, lungs, and kidneys among the five groups.

**Figure 3 ijms-23-06187-f003:**
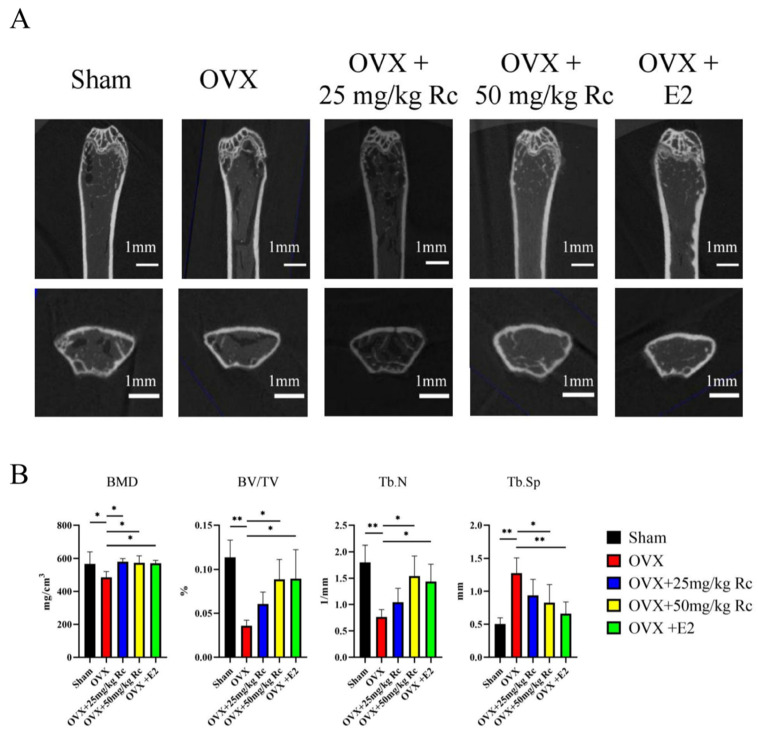
Ginsenoside Rc can increase BMD in OVX mice. (**A**) Three-dimensional reconstructed images of micro-CT 3 months after surgery. (**B**) Calculation of BMD, BV/TV, Tb.N and Tb.Sp using micro-CT 3 months after surgery. Data are presented as the mean ± SD, *n* = 3 specimens/group, * *p* < 0.05, ** *p* < 0.01. BMD, bone mineral density; BV/TV, bone volumetric fraction; Tb.Sp, trabecular separation; Tb.N, trabecular number.

**Figure 4 ijms-23-06187-f004:**
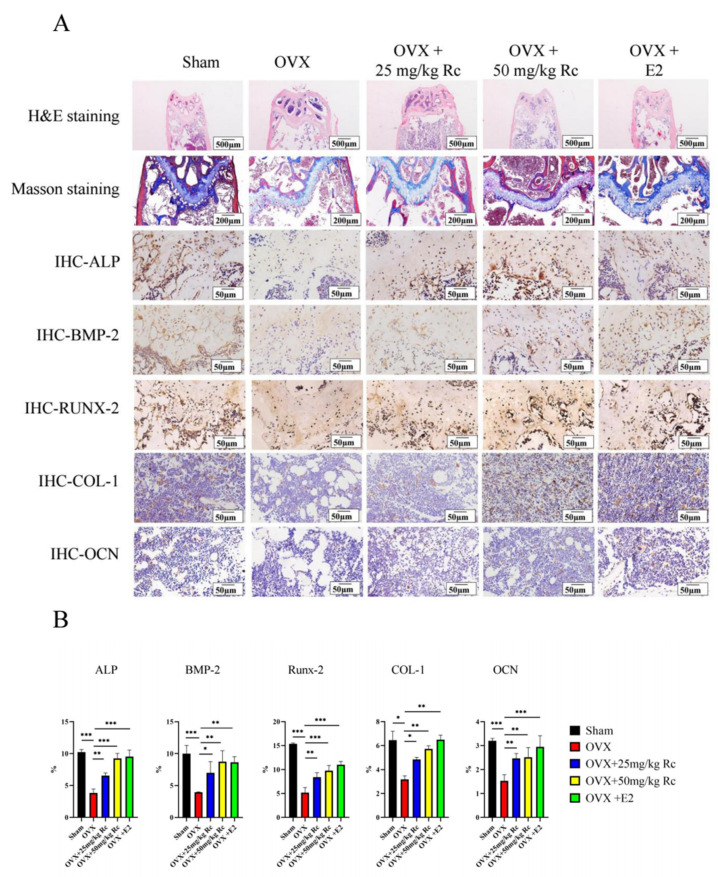
Effects of ginsenoside Rc on trabecular bone tissue of distal femur in mice. (**A**) Histological analysis of the distal femur including HE staining, Masson staining, and immunohistochemistry. (**B**) The expression of ALP, BMP-2, RUNX-2, COL-1, and OCN in stained femurs were analyzed using Image J software. Data are presented as the mean ± SD, *n* = 3 specimens/group, * *p* < 0.05, ** *p* < 0.01, *** *p* < 0.001.

**Figure 5 ijms-23-06187-f005:**
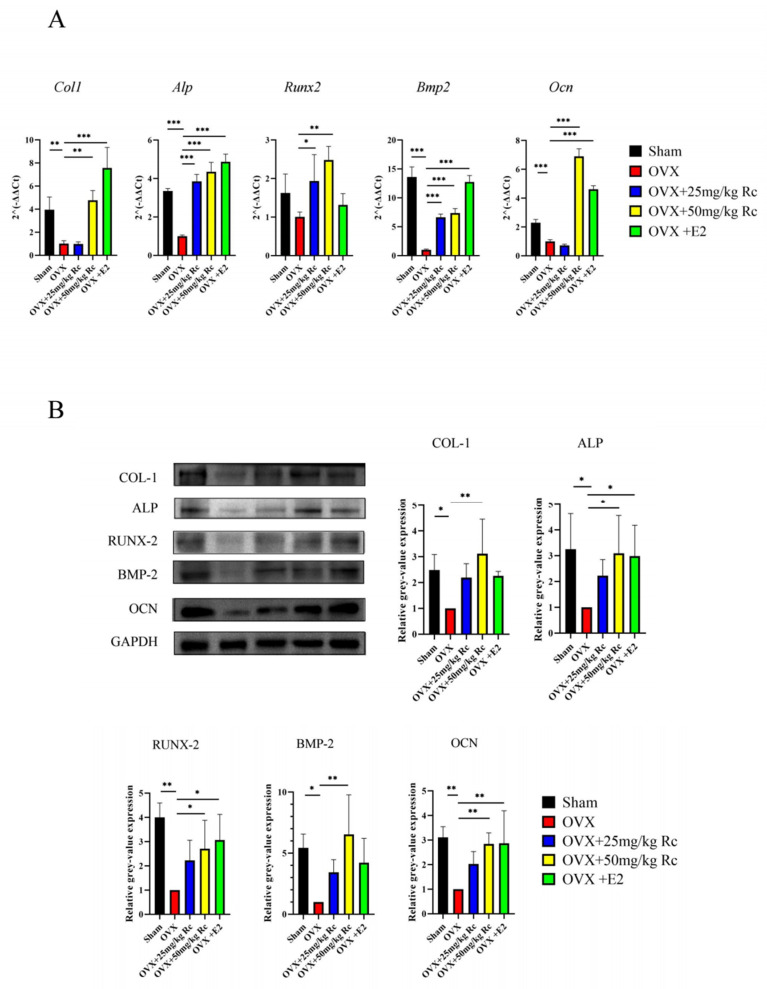
Ginsenoside Rc can prevent OVX-induced bone loss by promoting the expression of bone formation-related genes. (**A**) After 14 days of osteogenic induction, the expression of *Col1*, *Alp*, *Runx2*, *Bmp2*, and *Ocn* mRNA was assessed via qRT-PCR. (**B**) The expression of *Col1, Alp, Runx2, Bmp2,* and *Ocn* was assessed via Western blotting analysis after 14 days of osteogenic induction. Quantification of protein expression is shown. Data are presented as the mean ± SD, *n* = 3 specimens/group, * *p* < 0.05, ** *p* < 0.01, *** *p* < 0.001, relative to the OVX group.

**Figure 6 ijms-23-06187-f006:**
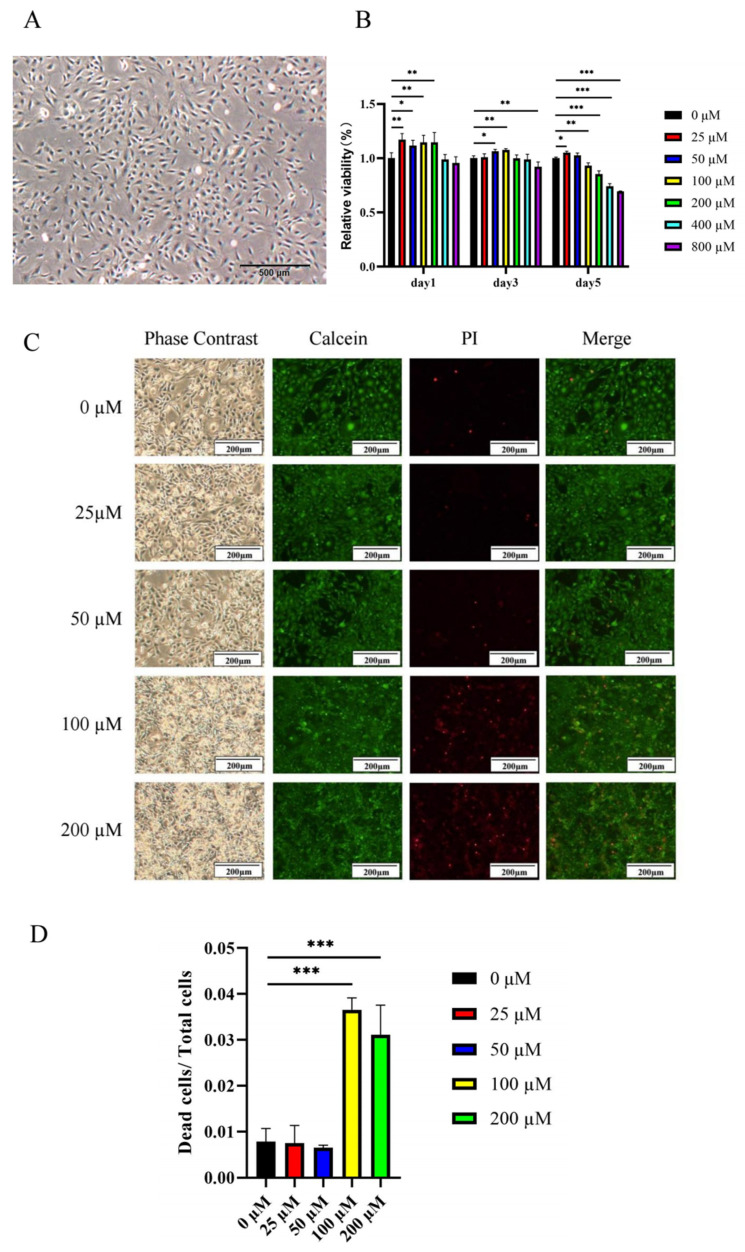
Low concentrations of ginsenoside Rc promote the viability of MC3T3-E1 cells. (**A**) Growth of MC3T3-E1. (**B**) The cell viability of MC3T3-E1 cells on days 1, 3, and 5. (**C**) Fluorescent image of live-dead cell staining of MC3T3-E1 cells on day 5. (**D**) The number of live-dead MC3T3-E1 cells on day 5. Data are presented as the mean ± SD, *n* = 3 specimens/group, * *p* < 0.05, ** *p* < 0.01, *** *p* < 0.001.

**Figure 7 ijms-23-06187-f007:**
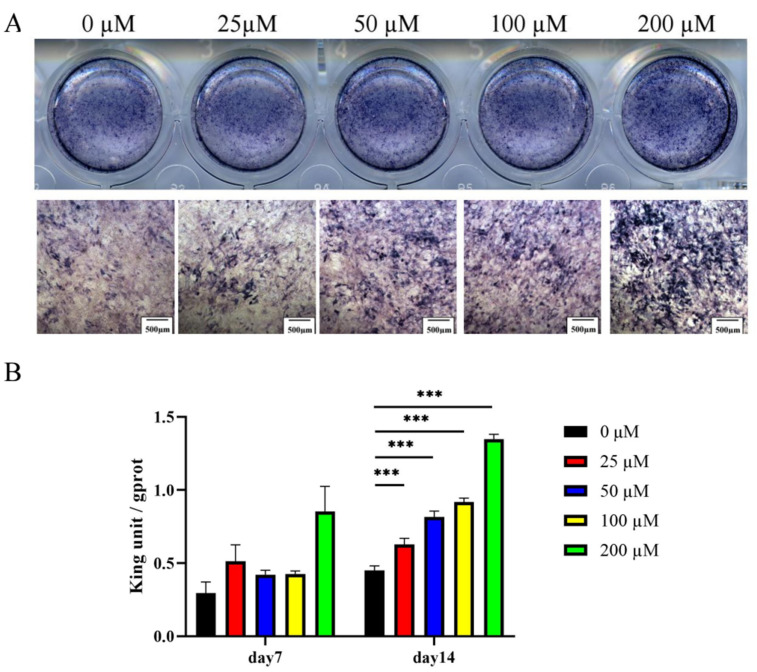
Ginsenoside Rc dose-dependently promotes the ALP activity in MC3T3-E1 cells. (**A**) ALP staining and (**B**) ALP quantitative assay on days 7 and 14 of osteogenic differentiation. Data are presented as the mean ± SD, *n* = 3 specimens/group, *** *p* < 0.001.

**Figure 8 ijms-23-06187-f008:**
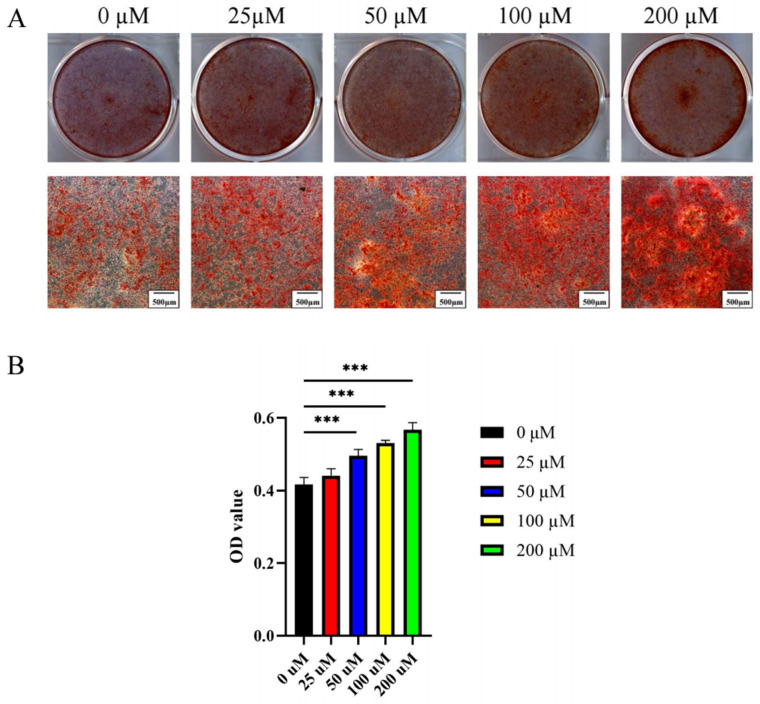
Ginsenoside Rc dose-dependently promotes the osteogenic mineralization of MC3T3-E1 cells. Mineralization was measured via (**A**) Alizarin Red staining and (**B**) quantitative assay after 21 days of osteogenesis. Data are presented as the mean ± SD, *n* = 3 specimens/group, *** *p* < 0.001.

**Figure 9 ijms-23-06187-f009:**
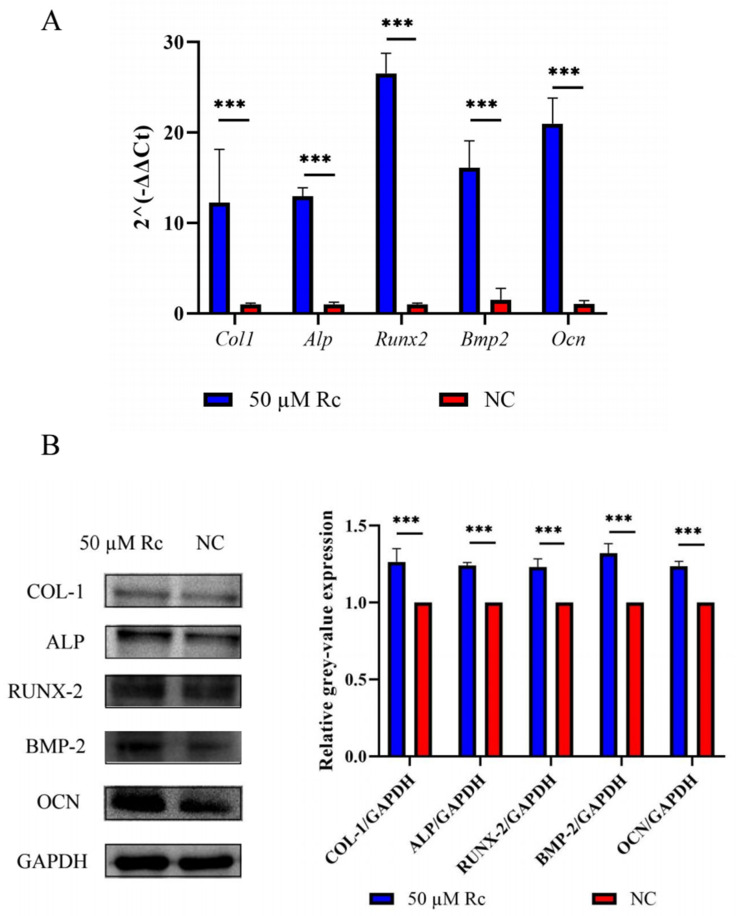
Ginsenoside Rc at 50 μM promotes the expression of osteogenesis-related genes. (**A**) The expression of *Col1*, *Alp*, *Runx2*, *Bmp2*, and *O*cn mRNA after 14 days of osteogenic induction with 50 μM ginsenoside Rc compared to control groups. (**B**) The expression of COL-1, ALP, RUNX-2, BMP-2, and OCN after 14 days of osteogenic induction with 50 μM ginsenoside Rc compared to the control groups, and the quantification of protein expression. Data are presented as the mean ± SD, *n* = 3 specimens/group, *** *p* < 0.001.

**Figure 10 ijms-23-06187-f010:**
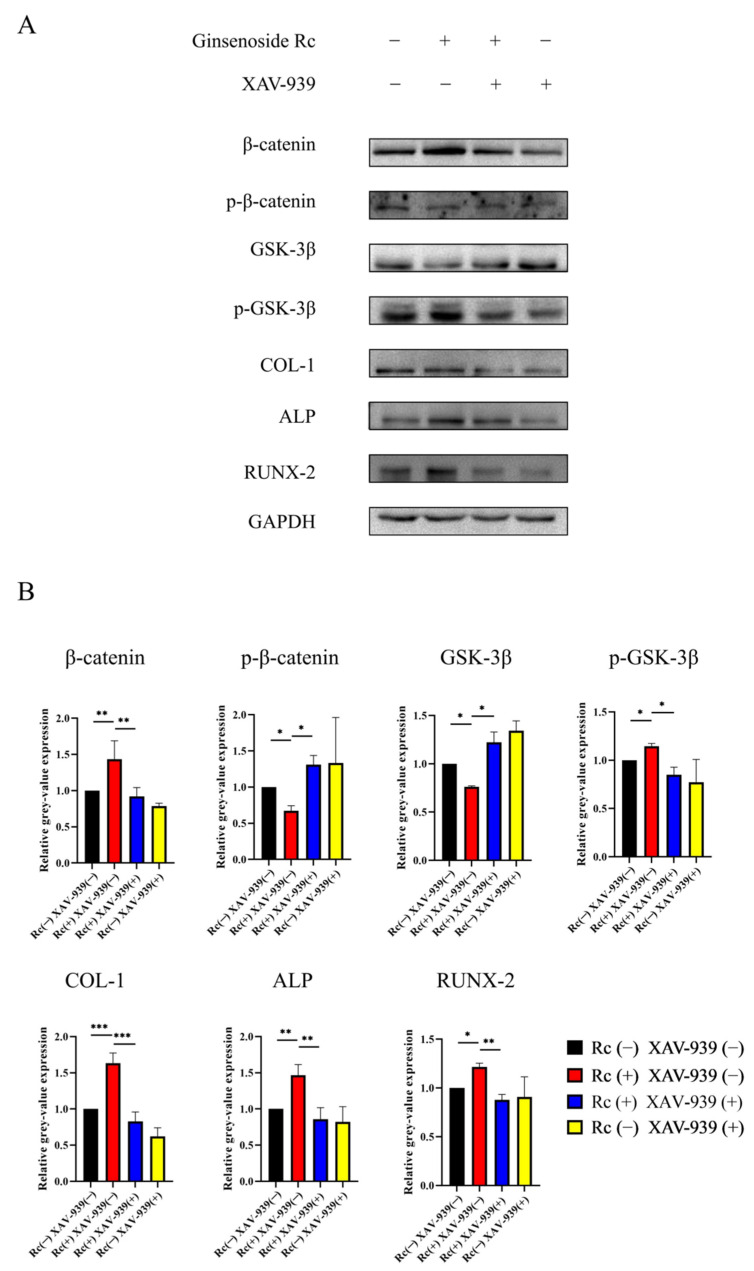
The expression of the Wnt/β-catenin pathway proteins in the cells after treatment with ginsenoside Rc or XAV-939. (**A**) Protein expression after 14 days was assessed via Western blotting. (**B**) Relative quantitative analysis of Western blotting results for β-catenin, p-β-catenin, GSK-3β, p-GSK-3β, COL-1, ALP, and RUNX2. Data are presented as the mean ± SD, *n* = 3 specimens/group, * *p* < 0.05, ** *p* < 0.01, *** *p* < 0.001.

**Figure 11 ijms-23-06187-f011:**
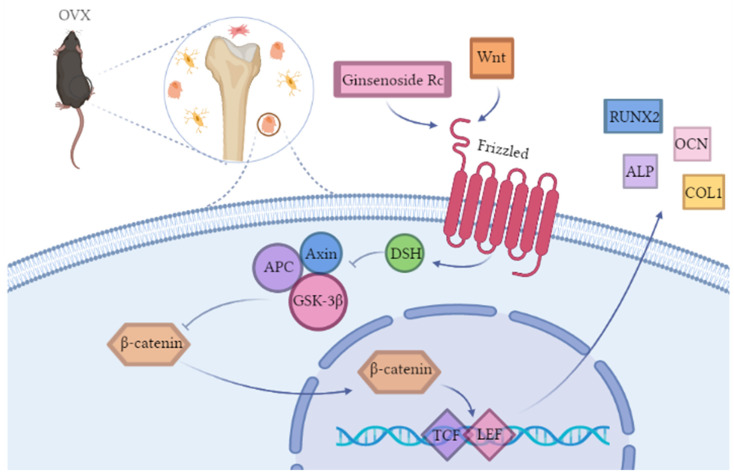
Schematic illustration of the role of ginsenoside Rc in the Wnt signaling pathway (created with BioRender.com). OVX: Ovariectomize; DSH: Dishevelled; APC: Adenomatosis Polposis Colis; GSK-3β: Glycogen synthase kinase 3β; TCF/LEF: T-Cell factor/Lymphoid Enhance Factor; RUNX2: Runt-related transcription factor 2; OCN:Osteocalcin; ALP: Alkaline phosphatase; COL1: Collagen I.

**Figure 12 ijms-23-06187-f012:**
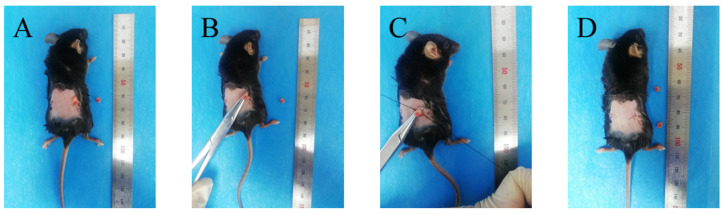
The surgical procedure of ovary removal. (**A**) Pink ovary and adipose tissue are exposed. (**B**) Ovaries are attached. (**C**) Ovaries are removed. (**D**) The skin and muscles are sutured in layers.

**Table 1 ijms-23-06187-t001:** Mouse primer sets used for qPCR

Genes	Primer Sequence, 5ʹ–3ʹ	
	Forward	Reverse
*Runx2*	GATGATGACACTGCCACCTCTGAC	TGAGGGATGAAATGCTTGGGAACTG
*Alp*	CGGCGTCCATGAGCAGAACTAC	CAGGCACAGTGGTCAAGGTTGG
*Bmp2*	AAGCGTCAAGCCAAACACAAACAG	GAGGTGCCACGATCCAGTCATTC
*Ocn*	CAGAGGAACTGGTTAGCAGGCAAC	ACGCAGGTTCTCAATGGCACAC
*Col1*	AGGGTCCCGCTGGTCAAGATG	ATGCCTGTTGCTGGTTCTGTAGTG

## Data Availability

Not applicable.

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
