# Peer review of "Ginsenoside Rc Promotes Bone Formation in Ovariectomy-Induced Osteoporosis In Vivo and Osteogenic Differentiation In Vitro"

_ijms, 2022, doi:10.3390/ijms23116187_

Round 1

Reviewer 1 Report

In this work Yang and coworkers tested the Ginsenoside Rc biological activity in regard to bone tissue mineralization during in vitro and in vivo experiments. Whereas the impact of Rc on bone mineralization was confirmed with ovariectomy mice, the molecular mechanism was determined during in vitro studies. The results are very important from the point of effect induced by dietary components. The manuscript is potentially interesting for the readers, but it needs to be revised by the Authors:

  • I suggest to modify the Abstract and instead of listing the methods used in the studies present more information about the obtained results;
  • Line 69 – present the full name of CK;
  • In the introduction present more details about the molecular mechanism with Wnt involvement in osteogenesis process and its importance;
  • In the introduction the aim of the study needs to be more emphasized in regard to the type of experimental models used – why were used C57Bl/6 female SPF mice or MC3T3-E1 cells;
  • Explain the rationale for usage of Rc doses - 25 or 50 mg/kg;
  • Please enlarge all the photos presented in the manuscript;
  • In Fig. 5 please add information that as the control the OVX were used;
  • For in vitro study I suggest to use unit µM instead of µmol/L;
  • The assay cell counting kit-8 (CCK-8) presents the results of Rc influence on cell metabolism, not cell proliferation and this needs to be modified within the manuscript;
  • Figure 6B – there is wrong unit presented; explain the method of calculation allowing presentation of data in Fig. 6D;
  • present detailed protocol of MC3T3-E1 cells incubation with Rc (medium removal, Rc addition period); why the incubation was no longer than 5 days?
  • The authors wrote that the Rc at 25- 50 µM had no cytotoxic activity on cells after 5 days of incubation; therefore explain why the Figure 7 presents data after cells incubation with 200 µM of Rc, and what is more 200 µM of Rc enhanced mineralization in cells after 21 days of incubation? Please explain with details the incubation time with the compounds – 1 day, 5 days, 14 days or 21 days? What “gprot” means in Figure 7C – the activity of ALP can be measured. Figure 7 can be modified and better ordered showing ALP photos and activity, than mineralization; why for mineralization 21 days of incubation were chosen, whereas other studies were performed after 14 days (ALP, expression of mRNA encoding selected genes or proteins)
  • Line 240 - Explain usage of XAV-939
  • Lines 284-284- sentence “However, the specific staining of these five osteogenesis-related genes was significantly restored upon ginsenoside Rc treatment” is related to the proteins detected with immunohistochemistry and needs to be modified
  • Lines 307-317 – this fragment represents rather the results, not discussion of the results; I suggest its modification or removal;
  • Lines 325-328 – there are no explanations what are Fz receptor, DSH, GSK-3, APC, TCF/LEF and what is their significance (some of these information can be presented in the Introduction)
  • Figure 10 – in capture present the explanation of short forms used;
  • In the discussion there is no data about other known effects of ginsenosides (and their dose/concentration) on bone metabolism or Wnt signalling pathway;
  • In conclusions please match the dose of Rc used in vivo with the concentration of Rc used in vitro – and discuss the results in regard to the Rc bioavailability
  • Part 4.1 – please add the dietary pattern for animals, as well as for the compounds treatment
  • Part 4.2 – add information of medium removal pattern, as well as the incubation time with the compounds (was the compound added for 14 days/21 days during differentiation?)
  • Part 4.7 – since CCK-8 determines metabolic activity, therefore using “proliferation” is inappropriate term; why cell viability was determined for the longest period equal 5 days? If so, does it mean that the Rc was added only for 5 days but the ALP and Alizarin red staining (also wb, qRTPCR) were performed at 14/21 days after cells treatment?

 In summary, I suggest the major revision.

Reviewer 2 Report

The authors set out to determine the effects of Ginsenoside Rc treatment within the murine OVX osteoporosis model. While the aim of the study is clearly defined and the general study outline is well set up, there are some significant changes to the work presented and therefore the manuscript need prior to be considered for publication. Those are outlined below:

Major points:

Introduction:

The introduction needs to be re-written as some statements are too generally phrased, or are even interpreted wrongly. Specifically,

1) The second paragraph needs to be removed or re-worded. The currently available drugs are very effective, yet they may cause side effects in some patients. Be more specific of what they are and how frequent if you want to keep this paragraph and cite the relevant literature from the clinical trials.

2) Third paragraph - there is no need to defent one drug over the others others - just name them and be precise on their mode of action. Here it is too bluntly stated that e.g. estrogen causes breast cancer. That is so not true - in what does, when and in how many cases. Rewording is necessary here.

3)"Traditional Chinese medicine 59
is famous for its multiple pharmacological effects and mild side effects." Has this ever been shown in clinical trials? If so, cite them here. If not, reword this paragraph too. Again, there is no need to defent one "drug" over another one, they can co-exist and the introduction should focus on the proven strengths and can mention the published weaknesses of each.

Methods:

In general those are not described enough in detail to allow for complete understanding of the performed work nor to allow for repetion. Specifically:

1) It is not clear how the immunohistochemical stainings were performed. The staining protocols and information on antibodies should be given. In addition, it is necessary to specify what and how quantifications of the stainings were performed. E.g. it is unclear to this reviewer why type I collagen expression in the bone marrow as supposed to the bone surface should have a direct implication on bone formation.

2) RTqPCR and Western Blots - please specify if the cortical shell or the bone without bone marrow or something different was utilized for these analyses.

Results:

1) While the OVX mouse model is an established model for estrogen-deficiency induced osteoporosis, it is mainly characterized by elevated bone resorption with no major changes to bone formation. It is therefore imperiable to determine osteoclast parameters by serum resorption markers and or bone histomorphometry with the Ginsenoside Rc treatment. 

2) The decription of the uCT results should be reworded. As no longitudinal study was done, words like "increased" should be avoided. In addition, the previous point is also of importance here, while the net bone loss could be partially prevented, it is unclear if this is due to elevated bone formation or reduced bone resorption or both. Please add analyses to reflect 1) and reword this paragraph.

3) Histomorphometry or serum bone formation analyses are lacking to support the finding of elevated bone formation due to Ginsenoside Rc treatment.

Discussion:

In general, the discussion needs to be shortened and adapted after the changes have been made to the results section.

Minor points:

Results:

1) The uCT images of the longitudinal cut from the femora should show the same plane. Please correct all to look like OVX+25mg RC.

Round 2

Reviewer 1 Report

The Authors answered to most of my questions and the mansucript has been improved. I suggest its acceptance for publication in IJMS.